# Nitrogen Fertilizer Induced Alterations in The Root Proteome of Two Rice Cultivars

**DOI:** 10.3390/ijms20153674

**Published:** 2019-07-26

**Authors:** Jichao Tang, Zhigui Sun, Qinghua Chen, Rebecca Njeri Damaris, Bilin Lu, Zhengrong Hu

**Affiliations:** 1Hubei Collaborative Innovation Center for Grain Industry, Agricultural College, Yangtze University, Jingzhou 434025, China; 2State Key Laboratory of Biocatalysis and Enzyme Engineering, School of Life Sciences, Hubei University, Wuhan 430062, China

**Keywords:** nitrogen fertilizer, rice, proteome, cultivars, nitrogen use efficiency (NUE)

## Abstract

Nitrogen (N) is an essential nutrient for plants and a key limiting factor of crop production. However, excessive application of N fertilizers and the low nitrogen use efficiency (NUE) have brought in severe damage to the environment. Therefore, improving NUE is urgent and critical for the reductions of N fertilizer pollution and production cost. In the present study, we investigated the effects of N nutrition on the growth and yield of the two rice (*Oryza sativa* L.) cultivars, conventional rice Huanghuazhan and indica hybrid rice Quanliangyou 681, which were grown at three levels of N fertilizer (including 135, 180 and 225 kg/hm^2^, labeled as N9, N12, N15, respectively). Then, a proteomic approach was employed in the roots of the two rice cultivars treated with N fertilizer at the level of N15. A total of 6728 proteins were identified, among which 6093 proteins were quantified, and 511 differentially expressed proteins were found in the two rice cultivars after N fertilizer treatment. These differentially expressed proteins were mainly involved in ammonium assimilation, amino acid metabolism, carbohydrate metabolism, lipid metabolism, signal transduction, energy production/regulation, material transport, and stress/defense response. Together, this study provides new insights into the regulatory mechanism of nitrogen fertilization in cereal crops.

## 1. Introduction

Nitrogen (N), an essential plant nutrient, is a key factor limiting the crop growth and productivity [1,2]. The amount of applied fertilizer in the world has increased 10-fold in the last half century, to meet the food demand of the growing world population. This increasing trend is predicted to be even greater in this century [3]. However, less than half of the field applied nitrogen fertilizer is absorbed and utilized by plants, while most of it is dissipated in the atmosphere, and leached into groundwater, lakes and rivers, which induce increasingly severe pollutions to the environment [4]. Thus, the improvement of N use efficiency (NUE) is an urgent task for agricultural sustainability and environmental protection. Rice (*Oryza sativa* L.), cultivated in more than 100 countries, is the major food source of at least half of the world’s population [5]. Therefore, proper management of N fertilizer is pivotal for improvement of rice yield.

Plants have developed an intricate N detection system, in which N status is constantly sensed and the plants continuously adapt to the changes by regulating gene expression, enzyme activity, and substance metabolism [6,7]. In plants, inorganic nitrogen is firstly reduced to ammonia (NH_3_) and then incorporated into organic compounds [8,9]. Afterwards, ammonia is assimilated into these amino acids that serve as vital nitrogen carriers in plants, including glutamine, glutamate, asparagine, and aspartate [10]. The major enzymes, responsible for the biosynthesis of these nitrogen-carrying amino acids, are glutamine synthetase (GS), glutamate dehydrogenase (GDH), glutamate synthase (GOGAT), aspartate aminotransferase (AspAT), as well as asparagine synthetase (AS) [10]. GS is a key enzyme for ammonia assimilation, which has high affinity for ammonia and ability to incorporate ammonia efficiently into organic combination (compounds). The glutamate synthase cycle has been well established in that NH_3_ is converted into organic compounds via its assimilation by GS, and this is the major route of nitrogen assimilation in plants [11]. In addition, N can be transferred into and out of proteins in different organs, and be moved between different organs in plants, which are achieved via the activity of transaminases, glutamine-amide transferases and GS [10]. However, excessive ammonium ion (NH4^+^) is one of the inorganic pollutants in aqueous solution, which could induce eutrophication and impair self-purification of water [12].

Numerous researches have been conducted to illuminate the regulatory mechanisms involved in NUE of crop plants [13]. Genetic engineering methods have become the research focus for improvement of biological N fixation [1,14,15,16]. However, these techniques are not very effective in improving NUE of crop plants. Furthermore, manipulating genes involved in the nitrogen uptake and assimilation have not yielded any obvious improvement of nitrogen utilization capacity of plant [1,14]. For example, overexpression of *nitrite reductase* (*NiR*) genes, which encodes the enzyme catalyzing the first and the rate-limiting step of N assimilation, did not cause any obvious change of NUE in plants [14].

Recently, the proteomic approach has become an effective tool in the study of nitrogen regulation. Ding et al. (2011) identified twelve protein spots by two-dimensional gel electrophoresis (2-DE), providing a better understanding of the mechanism underlying rice response to low nitrogen stress [17]. Besides, Liao et al. (2012) demonstrated that 47 differentially expressed proteins were identified in young ears of maize treated with different levels of nitrogen fertilizer, and the regulatory process was associated with hormonal metabolism [2]. Chandna and Ahmad (2015) investigated the effects of the nitrogen nutrition on protein expression pattern of leaves in low-N sensitive and low-N tolerant wheat (*Triticum aestivum* L.) varieties, and found proteins related to photosynthesis, glycolysis, nitrogenmetabolism, sulphur metabolism and defense [18]. Hakeem et al. (2012) analyzed the leaf proteins expression patterns of high and low N-responsive contrasting rice genotypes grown at three levels of N fertilizer, and determined that the differentially expressed proteins were involved in the energy production/regulation and metabolism [1].

Root system is essential and critical for plant growth, due to its involvement in nutrients and water absorption, as well as plant hormones synthesis [19,20]. Development of root system is remarkably affected by nitrogen supply and its distribution in the soil [21]. Liu et al. (2018) reported that five elite super hybrid rice cultivars displayed greater root traits and higher grain yield under N treatment at level of 300 kg/hm^2^ [22]. Total root number and total root length of these cultivars reached maximum at 55 days after transplanting. However, the molecular mechanism underlying N fertilizer response of root system is still unclear. In the present study, the agronomic traits of two rice cultivars were determined, conventional rice Huanghuazhan and indica hybrid rice Quanliangyou 681, grown at three levels of nitrogen fertilizer (including 135, 180 and 225 kg/hm^2^, labeled as N9, N12, N15, respectively). Then, the root proteins of the two rice cultivars under N nutrition (at the level of N15) were identified by using tandem mass tags (TMT)-labelling proteomic platform. This work will provide new insights into the N fertilizer response of rice root system, and the identified key proteins can be used for developing new strategies to improve NUE of rice.

## 2. Results

### 2.1. Determination of Agronomic Traits

Accumulation of dry matter is essential for crop yield formation, and dry weight is an extensively used parameter for assessing the growth conditions of plants. As shown in Table 1, the shoot dry weight increased with the growth of plants, which was also significantly affected by nitrogen fertilizer. For Huanghuazhan cultivar, three levels of N fertilizer treatments increased the values of dry weight at the four stages, and the positive effects of N12 and N15 were relatively outstanding. Similar results were observed in Quanliangyou 681, and N15 displayed the most remarkable impact. When compared, the difference in dry weight of the two rice cultivars showed that the values for Quanliangyou 681 were significantly higher than that of Huanghuazhan at a given level of N fertilizer at any stage. Cereal yield, a key agronomic trait for agricultural production, was also measured in this study. As shown in Figure 1, three levels of nitrogen fertilizer treatments increased the yields of Huanghuazhan cultivar in undifferentiated ways. Differentially, the production of Quanliangyou 681 improved with the increase of nitrogen fertilizer concentration, and the effect of N15 was the most notable. Furthermore, the yield of Quanliangyou 681 was significantly higher than that of Huanghuazhan with or without the treatment of nitrogen fertilizer (at the same level). These results suggested that the three levels of N fertilizer treatments could improve the growth and yields of the two rice cultivars, especially for Quanliangyou 681; and N12 and N15 could be used as the optimal concentration of nitrogen fertilizer treatment for Huanghuazhan and Quanliangyou 681, respectively.

### 2.2. Proteome-Wide Analysis of Differentially Expressed Protein of The Two Rice Cultivars Treated With Nitrogen Fertilizer

In total, 6728 proteins were identified by searching the rice database, with 6093 protein being quantified. All the annotation and quantification information of these proteins are presented in Appendix A. The quantitative repeatability of proteins was evaluated by principal component analysis (PCA) and relative standard deviation (RSD) statistical analysis. Good quantitative repeatability of proteins was determined by the small RSD values and high aggregation degree between repeated samples (Appendix A).

In this present study, proteins with the threshold change fold >2 or <0.5, and *p* value < 0.05 were considered as up-regulated and down-regulated proteins, respectively. As shown in Figure 2, the number of differentially expressed proteins and an overlap of these proteins were summarized. In total, 511 differentially expressed proteins in the two rice cultivars after application of N fertilizer were identified, and all the protein information is provided in Appendix A. N15 treatment induced 109 differentially expressed proteins (93 up-regulated and 16 down-regulated) in Quanliangyou 681, while 283 differentially expressed proteins (160 up-regulated and 123 down-regulated) in Huanghuazhan (Figure 2A). Under control condition, 68 up-regulated and 108 down-regulated proteins were identified in Huanghuazhan, when compared with Quanliangyou 681. After application of nitrogen, the expression levels of 63 proteins were increased and 115 proteins decreased in Huanghuazhan when compared with Quanliangyou 681 (Figure 2A). Figure 2B showed that there were 40 differentially expressed proteins shared by the comparisons of 681_N15 vs. 681_cK, and H_N15 vs. H_CK. Interestingly, 43 proteins revealed differentially expression abundance in two sample comparisons (H_CK vs. 681_CK, H_N15 vs. 681_N15) were regulated by rice cultivar difference, regardless of N fertilizer treatment.

### 2.3. Validation of Differentially Expressed Proteins by Parallel Reaction Monitoring

To validate the reliability of TMT-labeling proteomic results, parallel reaction monitoring (PRM) analysis was carried out. In detail, we selected 16 differentially expressed proteins due to their functional significance concluded from proteome analysis, and the ratio of protein abundance varied in a wide range. Then, we analyzed the expression levels of these 16 proteins among the four regimes (H_CK; 681_CK; H_N15; 681_N15). Consequently, a total of 64 quantitative values produced by PRM analysis were obtained. Correlation analysis was carried out of the 64 PRM values with their corresponding protein expression level generated by TMT labeling proteomic platform. As shown in Figure 3, the results generated by PRM analysis showed a strong correlation with the data produced by TMT labeling detection system (Pearson correlation coefficients r^2^ = 0.93). In addition, the distributions of peptide fragment ion peak area of all the 16 proteins is provided in the Appendix A.

### 2.4. Gene Ontology Classification Analysis of Differentially Expressed Proteins

To further explore the cellular functions of these differentially expressed proteins regulated by N fertilizer in rice, Gene Ontology (GO) classification analysis was performed. The top three cellular components were cell, membrane and macromolecular complex related proteins in Huanghuazhan, while extracellular region, cell, membrane related proteins in Quanliangyou 681 (Figure 4A). In the ontology of molecular function, binding and catalytic activity related proteins were the overwhelming preponderant in the two cultivars (Figure 4B). As for biological processes classification, metabolic process, single-organism process and cellular process related proteins were the dominant components in the two rice cultivars after N fertilizer treatment (Figure 4C).

### 2.5. Clustering Analysis of The Differentially Expressed Proteins

To better understand the involved pathways of these differentially expressed proteins, clustering analyses based on GO enrichments and Encyclopedia of Genes and Genomes (KEGG) pathway enrichments were conducted. Several biological processes altered in the two rice cultivars after application of nitrogen fertilizer, such as defense response to bacterium, fungus, and other organism; photosynthetic electron transport chain, as well as hexose metabolic process (Appendix A). As shown in Figure 5, the differentially expressed proteins in the comparison of 681_N15 vs. 681_CK were mainly enriched in valine, leucine and isoleucine degradation, amino sugar and nucleotide sugar metabolism, alpha-linolenic acid metabolism, galactose metabolism, cysteine and methionine metabolism. As for Huanghuazhan, the proteins regulated by nitrogen fertilizer were enriched in diverse pathways, such as photosynthesis, diterpenoid biosynthesis, alanine, aspartate and glutamate metabolism, linoleic acid metabolism, carbon fixation in photosynthetic organisms, nitrogen metabolism, arginine biosynthesis, and tryptophan metabolism. Under control condition, the differentially expressed proteins between the two rice cultivars were largely enriched in several pathways, including ribosome, photosynthesis-antenna proteins, plant-pathogen interaction, MAPK signaling pathway-plant. However, after treatment of the N fertilizer, the pathways enriched by differentially expressed proteins were changed, which mainly involved in biosynthesis of secondary metabolites, glycine, serine and threonine metabolism, phenylpropanoid biosynthesis, terpenoid backbone biosynthesis.

To obtain a better understanding of the possible regulatory mechanism of nitrogen fertilizer in different rice cultivars, several typical groups of differentially expressed proteins were summarized (Table 2, Table 3 and Table 4, and Appendix A). These proteins are involved in ammonium assimilation, signal transduction, substance metabolism, energy metabolism, material transport, and stress/defense response.

As shown in Table 2, several differentially expressed proteins involved in ammonium assimilation were identified in the comparisons of H_N15 vs. H_CK, and 681_N15 vs. 681_CK. As for Huanghuazhan cultivar, two glutamine synthetases (B8AVN6; B8AJN3) and one glutamate synthase (B8AWG6) were down-regulated, but one glutathione synthetase (B8AZE9) and one glutamate carboxypeptidase (B8A9D2) were found as up-regulated. Differentially, in Quanliangyou 681 cultivar, there was only one up-regulated glutamine synthetase (B8AQE3) identified.

In the present study, members of proteins associated with signal transduction were identified in in the comparisons of H_N15 vs. H_CK, and 681_N15 vs. 681_CK (Table 3). Two up-regulated receptor-like kinases (A2Z8U1; A2Y361) and two 1-aminocyclopropane-1-carboxylate oxidases (A2ZC94; B8AY41) were up-regulated, while one serine/threonine-protein kinase STY46 (B8B2K2) and one aspartokinase 1 (B8ANP3) were down-regulated by nitrogen fertilizer in Huanghuazhan cultivar. For Quanliangyou 681, all the proteins involved in signal transduction were up-regulated by N nutrition, including one cysteine-rich receptor-like protein kinase (B8B6Z1), two 1-aminocyclopropane-1-carboxylates (A2ZC94; B8AY41), one serine/threonine-protein kinase STY46 (B8AGJ3), and one phosphoenolpyruvate carboxykinase (A2XEP1); except aspartokinase 1 (B8ANP3).

Moreover, many differentially expressed proteins involved in energy metabolism and material transport were also found in the comparisons of H_N15 vs. H_CK, and 681_N15 vs. 681_CK. As shown in Table 4, one NAD(P)H-dependent oxidoreductase 2 (B8BFL3), and one glutamate dehydrogenase (A2XW22) were up-regulated in Quanliangyou 681 after treatment of N15, and similar results were observed in the category of material transport (one sugar carrier protein C, A2YML7; and one calmodulin-like protein, B8ACJ8). Differently, more proteins were found in Huanghuazhan cultivar with involvement in energy metabolism and material transport, such as NAD(P)H-dependent oxidoreductase 1/2, ferredoxin, ferredoxin--NADP reductase, some dehydrogenases, ABC transporter G family members, pyrophosphate-energized vacuolar membranes, potassium channel KAT2 isoform X1, transmembrane protein, etc.

Many proteins associated with substance metabolism were identified in the comparisons of H_N15 vs. H_CK, and 681_N15 vs. 681_CK (Appendix A), which included aminotransferase, methyltransferase, aspartic proteinase nepenthesin, glycosyltransferase, glucosidase, pyrophosphate--fructose 6-phosphate, lipoxygenase, etc. In the category of substance metabolism, the N fertilizer regulated proteins were more in the comparison of H_N15 vs. H_CK than that of 681_N15 vs. 681_CK. Although the number of differentially expressed proteins was smaller in Quanliangyou 681, all the proteins were up-regulated by N fertilizer in this cultivar, expect one O-methyltransferase 2 (Appendix A).

Proteins related to stress/defense response were also found to be regulated by N fertilizer, such as glutathione S-transferase, peroxidase, dehydrin, late embryogenesis abundant protein (LEA), momilactone A synthase, and chitinase, etc. Similarly, the differentially expressed proteins in the comparison of 681_N15 vs. 681_CK were less than that of H_N15 vs. H_CK, but most of them were up-regulated by N nutrition (Appendix A). Furthermore, the differentially expressed proteins in the comparisons of H_CK vs. 681_CK, and H_N15 vs. 681_N15 were also summarized in the above classifications (Appendix A). And the results suggested that the most proteins were down-regulated in Huanghuazhan compared with Quanliangyou 681, regardless of nitrogen fertilizer or not. Based on these results, we speculated that application of nitrogen fertilizer could influence a wide variety of biological processes in plants, and the expression levels of these proteins are relatively higher for Quanliangyou 681.

## 3. Discussion

To improve cereal production, application of inorganic nitrogenous fertilizers is one of necessary input. The amount of N fertilizer applied in agriculture has increased at an amazing speed in the last 50 years [12]. Consequently, the use of nitrogen fertilizers has already shown severe damage to the environment [1]. Therefore, it is urgent and crucial for the agricultural industry to analyze the molecular mechanism underlying NUE of crop plants. In the present study, we designed three levels for nitrogen fertilizer treatment of the two rice cultivars, including 135, 180 and 225 kg/hm^2^, labeled as N9, N12, N15, respectively. The results showed that three levels of N treatments increased the shoot dry weight and yields of the two rice cultivars, especially for Quanliangyou 681; N12 and N15 could be used as optimal concentration of N fertilizer application for Huanghuazhan and Quanliangyou 681, respectively (Table 1 and Figure 1).

Although previous studies have involved proteomic analysis of N fertilizer response in rice, most of them are concentrated on leaf proteins and are conducted by two-dimensional gel electrophoresis (2-DE) [1,3,23,24]. Studies focused on the changes of root proteome in response to N nutrition are limited. Ding et al. (2011) performed a proteomic study to investigate the response of rice root to low N stress, and they identified twelve proteins that were involved in tricarboxylic acid cycle, adenylate metabolism, phenylpropanoid metabolism, and protein degradation [17]. Besides, to analyze proteins regulated by nitrogen and cytokinin in rice roots, Ding et al. (2012) carried out a comparative proteomic analysis using two-dimensional polyacrylamide gel electrophoresis. Twenty-eight proteins were successfully identified, which were categorized into classes related to energy, metabolism, disease/defense, protein degradation, signal transduction, transposons, and unclear classification [25]. However, the number of identified differentially expressed proteins is relatively limited when using two-dimensional gel electrophoresis approach.

In the present study, a proteomic analysis was performed using TMT labeling detection platform, to explore the possible mechanism underlying N fertilizer response of rice root system. A total of 6093 proteins were quantified, and 511 differentially expressed proteins were identified (Figure 2 and Appendix A). Go analysis showed that metabolic process related proteins were the dominant group in the ontology of biological process (Appendix A). Consistently, the KEGG pathway analysis indicated that the proteins involved in amino acid metabolism, nitrogen metabolism and glycometabolism were significantly enriched in the two rice cultivars after treatment of N15 (Figure 5). These results determined that diverse biological processes may be involved in N fertilizer response of rice root system, which are summarized as follows.

### 3.1. Ammonium Assimilation

Plants take up inorganic nitrogen in the form of nitrate and ammonium from the soil and dinitrogen in the atmosphere. Ammonium is the final form of inorganic nitrogen, which is derived from primary nitrate reduction catalyzed by nitrate reductase and ferredoxin (Fd)-nitrite reductase, as well as absorption from soil and symbiotic dinitrogen fixation in root nodules of leguminous plants. Ammonium is then assimilated into glutamine and glutamate via glutamine synthetase (GS) and glutamate synthase (GOGAT) through GS/GOGAT cycle [26,27]. The glutamate (Glu) with ATP-dependent NH4^+^ are converted into glutamine (Gln) by GS, then the amide group of Gln is transferred to α-ketoglutarate (2-OG) to yield Glu [28]. It has been reported that the entire N in plant is channeled through the GS catalyzed reactions [28]. N atom can pass through the reaction catalyzed by GS many times, from uptake from the soil, assimilation and remobilization to final deposition in a storage protein of seed [29]. During this process, Gln serves as one important nitrogen donors for the biosynthesis of organic nitrogenous compounds (eg., nucleotides, amino acids and chlorophyll). Therefore, GS is a pivotal factor that controls nitrogen assimilation in plant. In this study, three differentially expressed GSs were identified, including two down-regulated (B8AJN3 and B8AVN6) in Huanghuazhan and one up-regulated (B8AQE3) in Quanliangyou 681 after treatment with N fertilizer (Table 2). Besides, one glutamate synthase 2 (B8AWG6) was decreased in Huanghuazhan, while one glutamate dehydrogenase (A2XW22) was increased in Quanliangyou 681. These results indicated that the roles of nitrogen fertilizer and the mode of nitrogen assimilation might be different between the two cultivars. Accordantly, Hakeem et al. (2012) revealed that the expression intensity of GS enhanced in rice genotype ‘Munga Phool’ (low nitrogen efficiency) after treatments with KNO_3_ at levels of 10 mM and 25 mM, but the intensity in ‘Rai Sudha’ (high nitrogen efficiency) decreased [1]. The vital role of GS has been highlighted in productivity of maize kernel in the study of Hirel and Lea (2001) by using a quantitative genetic approach [28]. One QTL for thousand kernels weight was identified coincident with a GS (Gln1–4) locus, while two QTLs for thousand kernel weight and yield were coincident with another GS (Gln1–3) locus. Furthermore, the positive correlation between kernel yield and GS activity has been revealed in previous study [30]. Herein, we deduced that the higher level of GS expression in Quanliangyou 681 could enhance the incorporation of NH4^+^ into organic compounds, which is beneficial to increase grain production.

### 3.2. Signal Transduction

It is very common for living organisms to perceive signal through cell-surface receptors [28]. In plants, several different types of cell-surface receptors are determined to perceive diverse signals and stimuli from the environment [31,32,33]. Plant receptor-like kinases (RLKs) are transmembrane proteins, defined by the presence of a signal sequence: an amino-terminal extracellular domain with a transmembrane region, and a carboxyl-terminal intracellular kinase domain [34,35]. RLKs control an extensive range of biological processes, including development, hormone perception, disease resistance and self-incompatibility. In our study, there were two receptor-like protein kinases up-regulated by N fertilizer application in Huanghuazhan (A2Z8U1; A2Y361), and one up-regulated in Quanliangyou 681 (B8B6Z1), indicating the enhancement of signal transduction by N fertilizer (Table 3). Besides, the expression levels of one serine/threonine-protein kinases STY46 (B8AGJ3) was increased in Quanliangyou 681, while that of (B8B2K2) was decreased in Huanghuazhan cultivar, after N fertilizer treatment, suggesting the difference in N-regulated signal transduction between the two cultivars. The synthesis of ethylene, one of classical plant hormones, is from methionine through S-adenosyl-L-methionine and 1-aminocyclopropane-1-carboxylic acid (ACC), which is catalyzed by ACC synthase and ACC oxidase, respectively [36]. Interestingly, two 1-aminocyclopropane-1-carboxylate oxidases (A2ZC94; B8AY41) were up-regulated in the two rice cultivars, which implied that ethylene signaling may be affected by N fertilizer. Based on these results, we can speculate that signal transduction may be involved in the nitrogen nutrition response of plants.

The 14-3-3 proteins play crucial roles in a variety of important physiological pathways regulated by phosphorylation. These proteins complete signal transduction by binding to the phosphorylated target, which accomplish an alteration of structure that regulates activity [33]. There are a wide range of processes controlled by 14-3-3s, including the fundamental nitrogen and carbon assimilation, starch synthase, Glu synthase, ATP synthase, ascorbate peroxidase, and methyl transferase [37,38]. In the present study, application of N fertilizer had no significant effect on the expression of 14-3-3 protein in the two rice cultivars; whereas the amount of five 14-3-3 proteins (A2XL95; A2XUA6; A2X6G8; A2YWB4; A2YVG3) were down-regulated in the comparison of H_CK vs. 681_CK (Appendix A). This result showed that the expression of 14-3-3 proteins may be of constitutive nature in the two rice cultivars, and the higher levels of these proteins in Quanliangyou 681 may be contributed to enhance signal transduction.

### 3.3. Energy Metabolism and Material Transport

Respiration is the process of organic matter oxygenolysis and energy production. Most oxidative degradation of respiratory substrates occurs through dehydrogenation reactions. Hydrogen acceptors, such as NAD+, FAD or NADP+, will be transferred into NADH, FADH2 or NADPH after accepting hydrogen ion and electron, which promotes synthesis of ATP to provide energy for organism [39]. As shown in Table 4, most of the proteins related to respiration were up-regulated by nitrogen fertilizer in Huanghuazhan cultivar, including NAD(P)H-dependent oxidoreductase (A2Z4G9; B8BFL3), ferredoxin (A2XNS0), dehydrogenase (A2ZCK1; B8ADR5; A2YBK1; A2XW22). As for Quanliangyou 681, one NAD(P)H-dependent oxidoreductase 2 (B8BFL3) and one glutamate dehydrogenase 2 (A2XW22) were also up-regulated after treatment of N fertilizer. These results indicated that application of nitrogen fertilizer could enhance energy metabolism in the two cultivars.

Membrane bound H^+^ pumps are proposed to constitute the primary transducers on the basis of the chemiosmotic hypothesis, which make living cells to interconvert light, chemical, and electrical energy [40]. Through the transmembrane electrochemical gradients, H^+^ pumps provide energy for the transport of other solutes, or transduce the H^+^ electrochemical gradient generated by membrane-linked anisotropic redox reactions to the synthesis of ATP [40]. The energy-dependent transport of solutes through vacuolar membrane of plants is driven by two kinds of H^+^ pumps: vacuolar (V type) H^+^-ATPase and H^+^-translocating (pyrophosphate-energized) inorganic pyrophosphatase (H^+^-PPase) [41]. Both H^+^-ATPase and H^+^-PPase are abundant and ubiquitous in the vacuolar membranes of plant cells, and they make an essential contribution to the H^+^-electrochemical potential difference. The ATP-binding cassette (ABC) transporter superfamily is the largest transporter gene family. The proteins bind ATP and utilize the energy to drive the transport of diverse molecules, including sugars, amino acids, peptides, proteins, metal ions, and plenty of hydrophobic compounds and metabolites across the plasma membrane and intracellular membranes [42,43,44]. In our study, N nutrition increased the expression levels of two ABC transporter G family members (B8BLE1; B8B6Q3) and one pyrophosphate-energized vacuolar membrane proton pump (A2WPG7) in Huanghuazhan, but no obvious change was observed in Quanliangyou 681. However, one sugar carrier protein C (A2YML7) and one calmodulin-like protein (B8ACJ8) were identified as up-regulated by N fertilizer in Quanliangyou 681(Table 4). These above results illustrated that energy metabolism and matter transport in plant cells might be associated with nitrogen nutrition response of plant root system.

### 3.4. Substance Metabolism

Aminotransferases have been extensively studied since their discovery over 80 years ago, owing to their vital functions in the synthesis of amino acids by catalyzing the process of amino transfer [45,46]. In the present study, three down-regulated (B8BGM4; B8APP5; B8AZ97) aminotransferases were identified in Huanghuazhan, and one up-regulated (A2XE78) was found in Quanliangyou 681, after application of nitrogen fertilizer (Appendix A). As a form of inorganic nitrogen, ammonium is derived from several metabolic pathways and assimilated into glutamine, glutamate, asparagine and carbamoylphosphate [27,47,48,49]. Specifically, the asparagine synthesis is catalyzed by asparagine synthetase through amidation of aspartate by using either glutamine or ammonium as an amino donor, which is catalyzed by ammonia-dependent or glutamine-dependent asparagine synthetase [50,51]. For most non-leguminous plants, glutamine and asparagine function as the major nitrogen transport and storage compounds from source to sink organs [52]. Due to the high nitrogen/carbon ratio and stability, asparagine is considered as an optimal nitrogen transport and reserve compound. Interestingly, the expression level of one glutamine-dependent asparagine synthetase (B8ALL8) was significantly increased by N fertilizer in Quanliangyou 681, while no obvious change was observed in Huanghuazhan cultivar. This result revealed the difference in nitrogen metabolism between the two rice cultivars, and the probable superiority of nitrogen transport and reserve for Quanliangyou 681.

Glycosidase, also known as glycosidic hydrolase, is a major category of hydrolases belonging to the glycosylase family [53]. They are the enzymes hydrolyzing glycosidic bonds, and playing a critical role in carbohydrate metabolism. Glycosyltransferases are divided into 47 different families based on both PSI-BLAST sequence analysis and substrate/product stereochemistry [54]. The glycosyltransferase reaction is the process involving the transfer of a monosaccharide from an activated sugar donor to a saccharide, protein, DNA, lipid or small molecule acceptor [55,56]. Glycosyltransferase-mediated reaction is proposed to proceed through an oxocarbenium-ion-like transition state, which is similar to glycosidase reactions [57,58]. In our study, two glycosyltransferase (A2Z7U4; A2WUT6) and 6 glycosidases (B8B1F4; B8B1F5; B8B6Z6; A2WYZ7; A2WYX6; A2WYX5) were significantly up-regulated in Huanghuazhan, while only two glycosidases (A2WYZ7; A2WYX6) were increased in Quanliangyou 681 cultivar, after treatment of N fertilizer (Appendix A). These results indicated that the enhancements of substance metabolism induced by N fertilizer are different between the two rice cultivars, and Huanghuazhan seems to be more sensitive to nitrogen nutrition.

### 3.5. Stress and Defense Response

Peroxidase (POD) is one key enzyme for the oxidative detoxification, due to its high affinity for H_2_O_2_ [59]. It has been determined that de-novo synthesis of the enzymatic protein or up-regulation of related gene, are contributed to the increase of antioxidant enzyme activity [60,61]. In our study, many PODs were changed after N fertilizer treatment in the two rice cultivars, with 5 up-regulated in Quanliangyou 681; while 4 up-regulated and 6 down-regulated in Haunghuazhan (Appendix A). Late embryogenesis abundant protein (LEA) protein has been proposed to function as an antioxidant and a membrane stabilizer during water stress, which plays an important role in plant abiotic stress tolerance [62]. LEA proteins are divided into several different groups, among which one group (LEA II) constituting of dehydrins (DHNs). DHNs have been reported to function as membrane stabilizers during freezing induced dehydration [63], functioning as possible osmoregulatory substance [64] or as radical scavengers [65]. Overexpression of *LEA*/*DHN* enhanced the stress tolerance in transgenic plants [66,67]. In the present study, three LEA proteins (A2YTZ6; A2Y720; A2WU85) and one dehydrin (A2ZDX4) were significantly increased in Quanliangyou 681 after N fertilizer treatment, but no obvious difference was observed in Huanghuazhan cultivar (Appendix A). Glutathione S-transferase acts as a detoxification enzyme that catalyzes the reduction of tripeptide glutathione (GSH; g-Glu- Cys-Gly) into multitudinous hydrophobic and electrophilic substrates [68]. It has been determined that GSTs are associated with high tolerance to abiotic stress in plant [68,69]. In this present study, 5 up-regulated (A2Z9M2; A2WZ34; A2Z9L3; A2Z9J9; A2Z6F5) and 2 down-regulated (A2XK19; A2WQ51) GSTs were found in Huanghuazhan after nitrogen fertilizer application (Appendix A). However, only one GST (A2WQ51) changed (down-regulated) in Quanliangyou 681 treated with N nutrition. These results indicated that the effects of nitrogen nutrition on antioxidant system are different between the two cultivars. Although the antioxidant enzymes in Hunaghuazhan seemed to be more sensitive to N fertilizer, their expression levels were relatively higher for Quanliangyou 681 (Appendix A, see the comparison of H_15 vs. 681_N15). Above these results, we speculated that the higher stress tolerance in Quanliangyou 681 may be beneficial for root absorbing the water and nutrition, consequently improving the growth and grain production.

Momilactone A and momilactone B are important phytoalexins in rice, which play vital roles in the rice defense system against pathogens and insects [70,71,72]. Chitinase also plays an important role in cell defense, by attacking on chitin molecules that are the main structural component in fungal cell wall and insects’ skeleton [18,73]. Besides this, dramatic increase in chitinase level by numerous abiotic agents (ethylene, salicylic acid, salt solutions, ozone, UV light) and by biotic factors (fungi, bacteria, viruses, viroids, fungal cell wall components and oligosaccharides) also proved their role in plant defense response [74,75]. In our present study, the expression levels of several momilactone A synthases (B8B5L2; B8B5L4; A2YPN5; A2YPP1; A2Z1W4; B8B5L1) and chitinases (A2Y4F6; A2Z7A3; A2Y4F5) were significantly increased by N fertilizer application in Huanghuazhan cultivar, whereas no obvious change was observed in Quanliangyou 681 (Appendix A). This result indicated that the defense system could be improved by N nutrition in Huanghuazhan cultivar.

## 4. Conclusions

This study highlighted the differentially expressed proteins in the two rice cultivars’ response to N nutrition. These proteins were mainly involved in ammonium assimilation, substance metabolism, signal transduction, energy metabolism, material transport, and stress/defense response. Although the number of N fertilizer-regulated proteins was more for Huanghuazhan cultivar, the expression levels of most key proteins were relatively higher in Quanliangyou 681. Therefore, we speculated that Huanghuazhan is more sensitive to N nutrition, while the higher yield improved by N fertilizer of Quanliangyou 681 may be owing to the relatively higher background levels of these responsive proteins. These proteins provide a better understanding of nitrogen regulatory functions in cereal crops, which is necessary for precise identification of potential molecular protein markers to assist the breeding for high NUE cultivars.

## 5. Materials and Methods

### 5.1. Plant Materials and Growth Conditions

Seeds of two rice cultivars, conventional rice Huanghuazhan and indica hybrid rice Quanliangyou 681, were used in this study, which were provided from Hubei Seed Industry Group and Hubei Winall Hi-tech Seed Company, respectively. This present experiment was performed at Lake Tai farm in Jinzhou city during the rice-growing season (May–October, 2017). The soil fertility was medium, and the previous crop was wheat. Seedlings of the two rice cultivars were treated with nitrogen fertilizer, at three levels of 135, 180 and 225 kg/hm^2^ (labeled as N9, N12, N15, respectively), and the untreated group was considered as the control (labeled as N0). For N fertilizer treatment, basal fertilizer was compound fertilizer (15:15:15); tillering fertilizer and panicle fertilizer were urea, and the proportion of the three N fertilizers was 6:2:2. Phosphate fertilizer was one-off basal application, and potash fertilizer was used as basal and panicle fertilizer with the ratio of 7:3. Totally, the ratio of fertilizing amounts of N, P, K was 2:1:2. The rice seeds were soaked, pre-germinated for 48h, and sown in soil. Every group had three biological replicates, and each of them was grown within an independent zone (30 m^2^), and different zones were isolated by ridges, with independent managements of water and fertilizer.

### 5.2. Measurement of Rice Dry Weight and Grain Yield

The plant samples for dry weight measurement were collected at four key stages, including tillering, jointing, full heading and mature period. For one zone, five holes of plants were collected without roots. The samples were dried at 105 °C for 30 mins followed by 80 °C for 48 h, the dried samples were used for measurement of shoot dry weight after being allowed to cool to room temperature. After harvesting, the grain yields of the two rice cultivars were determined, respectively.

### 5.3. Protein Extraction

Four regimes were designed in the proteomic study, and each regime had three biological replicates. The groups of H_CK, and 681_CK represented as Huanghuazhan or Quanliangyou 681 under control condition; while H_N15 and 681_N15 indicated as Huanghuazhan or Quanlaingyou 681 under N treatment, respectively. The root samples in rice booting stage were collected respectively, and frozen in liquid nitrogen, then kept at −80 °C. Protein extraction was determined by the method described by Hu et al. (2017) and Zhou et al. (2018) with some modifications [76,77]. The sample was ground into powder, then transferred into extraction buffer (containing 10 mM dithiothretiol, 1% protease inhibitor and 2 mM EDTA), and sonicated three times by using a high intensity ultrasonic processor (Scientz). Isometric Tris-saturated phenol (pH 8.0) was added, and the mixture was centrifuged at 5000× *g* at 4 °C for 10 min, the supernatant was transferred into a new centrifuge tube. Proteins were added five volumes of ammonium acetate/ methanol and precipitated for 12 h. The remaining precipitate was washed with ice-cold methanol and acetone, successively. Finally, the protein was redissolved in 8 M urea. The protein concentration was measured by the method described by Walker (2009) [78], by using BCA kit according to the manufacturer’s instructions.

### 5.4. Trypsin Digestion

Trypsin digestion was determined according to the method described in our previous study with slight modifications [76]. In detail, the protein solution was added with 5 mM dithiothreitol (DDT) and reduced at 56 °C for 30 min. Then the solution was alkylated with 11 mM iodoacetamide for 15 min at room temperature in darkness. The protein sample was diluted by adding 100 mM triethylammonium bicarbonate (TEAB) to insure the urea concentration less than 2M. Finally, trypsin was added at the ratio of 1:50 (trypsin-to-protein mass ratio) for the first digestion at 37 °C overnight, and 1:100 for a second 4 h-digestion.

### 5.5. TMT Labeling

The tryptic peptide was desalted by Strata X C18 column (Phenomenex, Torrance, CA, USA) and vacuum freeze-dried. Peptide was reconstituted in 0.5 M TEAB and labeled by using TMT kit according to the manufacturer’s protocol. In brief, one-unit TMT reagent was dissolved in acetonitrile, and mixed with peptides, then incubated for 2 h at room temperature. The labeled peptide was pooled, desalted and vacuum freeze-dried.

### 5.6. HPLC Fractionation

The labeled peptides were fractionated by high pH reverse-phase HPLC with Agilent 300Extend C18 column (5 μm particles, 4.6 mm ID, 250 mm length). Briefly, peptides were firstly separated with a gradient of 8–32% acetonitrile (pH 9.0) over 60 min into 60 fractions. The peptides were combined into 18 fractions (final fraction 1 = 1, 19, 37, 55; final fraction 2 = 2, 20, 38, 56; final fraction 3 = 3, 21, 39, 57, ………; final fraction 18 = 18, 36, 54); afterwards, they were dried by vacuum centrifuging.

### 5.7. LC-MS/MS Analysis

The peptides were dissolved in solvent A, separated through the EASY-nLC 1000 UPLC system. The solvent A was composed of 0.1% formic acid and 2% acetonitrile, while solvent B contained 0.1% formic acid and 90% acetonitrile. The gradient was comprised of an increase from 9% to 25% solvent B over 26 min, 25% to 36% in 26–34 min, and climbing to 80% in 3 min and then holding at 80% for the last 3 min, all at a constant flow rate of 700 nL/min. The peptides, separated by UPLC system, were subjected into NSI ion source and analyzed with application of tandem mass spectrometry (MS/MS) in Q ExactiveTM Plus (Thermo, Bremen, Germany). The electrospray voltage applied was 2.0 kV. The m/z scan range was 350 to 1550 for full scan, and intact peptides were detected in the Orbitrap at a resolution of 60,000. Peptides were then selected for MS/MS using NCE setting as 100 and the fragments were detected in the Orbitrap at a resolution of 15,000. Data was collected by using a data-dependent procedure (DDA) that alternated between one MS scan followed by 20 MS/MS scans with 30s dynamic exclusion. The parameters of automatic gain control (AGC), signal threshold and maximum time of injection were set as 5E4, 5000 ions/s and 200 ms, respectively.

### 5.8. Database Search

The MS/MS data were further processed by using Maxquant search engine (v.1.5.2.8). Tandem mass spectra were searched against UniProt Oryza sativa_india database (37,383 sequences) coupled with reverse decoy database. Trypsin/P was designated as the specific cleavage enzyme allowing up to 2 missing cleavages. In addition, the mass tolerance for precursor ions was set as 20 ppm in First search and 5 ppm in Main search, and the mass error was set as 0.02 Da for fragment ions. Carbamidomethyl on cysteine was designated as fixed modification, while oxidation on methionine was specified as variable modifications. The quantitative method was set as TMT-6plex, and the false discovery rate (FDR) for the identification of protein and propensity score matching (PSM) were both adjusted to <1%.

### 5.9. Bioinformatics Methods

The bioinformatics analysis methods were according to the detailed description in the previous studies [76,77]. Totally, in this present study we performed the protein annotation, including Gene Ontology (GO) annotation, domain annotation, Encyclopedia of Genes and Genomes (KEGG) pathway annotation; functional enrichment analysis, including enrichment of GO analysis, pathway analysis and protein domain analysis; as well as enrichment-based clustering analysis.

### 5.10. Parallel Reaction Monitoring (PRM) Validation of Differentially Expressed Proteins

To validate the reliability of the results produced by TMT labeling proteomic platform, parallel reaction monitoring (PRM) analysis was carried out, which has emerged as an alternative method of targeted proteins/peptides quantification [79,80]. A total of 16 differentially expressed proteins were selected for the project of PRM, due to their functional significance concluded from proteome analysis, and the ratio of protein abundance varied in a wide range. The experimental operation was as the following: protein extraction and trypsin digestion were consistent with the above TMT-labelling detection proteomics analysis. As for separation of peptide fragment by an EASY-nLC 1000 UPLC system, the tryptic peptides were dissolved in 0.1% formic acid (solvent A). The gradient was comprised of an increase from 4% to 16% solvent B (0.1% formic acid and 90% acetonitrile) over 38 min, 16% to 30% in 14 min and climbing to 80% in 4 min then holding at 80% for the last 4 min, with a constant flow rate of 500 nL/min.

The peptides were subjected to NSI source followed by MS/MS in Q ExactiveTM Plus (Thermo, Bremen, Germany). The electrospray voltage was set as 2.0 kV. The m/z scan range applied was 400 to 1100 for full scan, and intact peptides were detected in the Orbitrap at a resolution of 70,000. Peptides were then selected for MS/MS with NCE setting as 27, and the fragments were detected in the Orbitrap with resolution of 17,500. A data-independent procedure was applied that alternated between one MS scan followed by 20 MS/MS scans. Automatic gain control (AGC) was set at 3E6 and 1E5 for full MS and MS/MS, respectively, while the maxumum IT was set at 50 ms for full MS and 300 ms for MS/MS. The isolation window for MS/MS was set at 1.6 *m*/*z*.

The resulting MS data was processed by using Skyline (v.3.6, Seattle, WA, USA). Parameters for peptide: enzyme was set as Trypsin [KR/P], and max missed cleavage set as 0. The peptide length was set as 7–25, and alkylation on cysteine was specified as fixed modification. Transition settings: precursor charges were set as 2 and 3; ion charges were set as 1; ion types were set as b, y. The fragment ions were selected from ion 3 to the last one, and the ion match tolerance was set as 0.02 Da.

### 5.11. Statistical Analysis

All experiments performed in this study were repeated three times. For analysis of agronomic characters, the values were shown as mean ± SD of three replicates, and statistical analyses were conducted by one-way analysis of variance (ANOVA) concatenated with independent-samples t test. For enrichments of GO analysis and KEGG pathway analysis, a two-tailed Fisher’s exact test was employed to test the enrichment of the differentially expressed protein against all identified proteins.

## Figures and Tables

**Figure 1 ijms-20-03674-f001:**
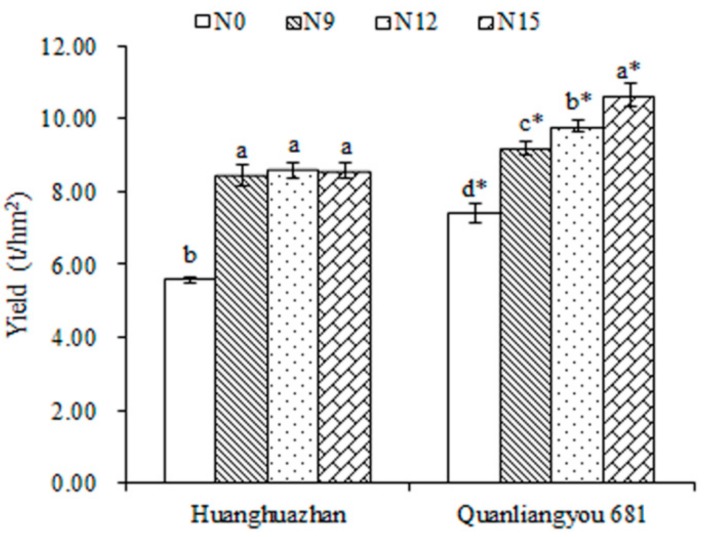
Grain yields of the two rice cultivars treated with N fertilizer at three levels. Seeds of two rice varieties, conventional rice Huanghuazhan and indica hybrid rice Quanliangyou 681, were soaked, pre-germinated for 48 h, and sown in soil. Seedlings of the two rice varieties were treated with nitrogen fertilizer at three levels (including 135, 180 and 225 kg/hm^2^, labeled as N9, N12, N15, respectively), and the untreated group was considered as the control (labeled as N0). Different letters (a, b, c) indicate significant differences among different nitrogen fertilizer levels for a given cultivar based on Duncan’s multiple range tests (*p* < 0.05). Star (*) represents statistical significance between the two cultivars at a given nitrogen fertilizer level based on Independent-samples *t* test (*p* < 0.05).

**Figure 2 ijms-20-03674-f002:**
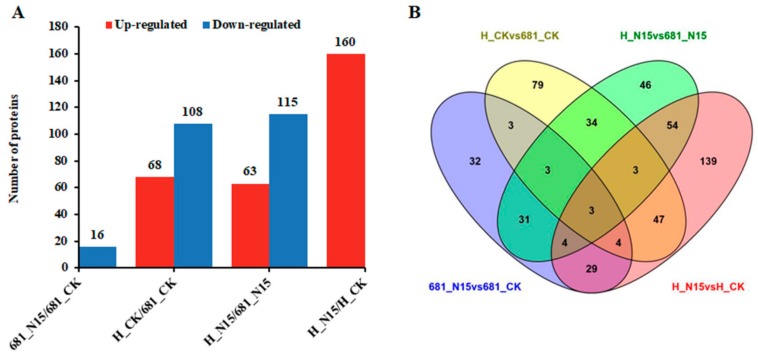
Histogram (**A**) and Venn diagram (**B**) of the number distribution of differentially expressed proteins in different comparison groups. Proteomics analysis was performed on the roots of the two rice cultivars at booting stage with/without treatment of N15 (225 kg/hm^2^ nitrogen fertilizer). N15 represented rice under nitrogen fertilizer treatment with the concentration of 225 kg/hm^2^, while control check (CK) represented the control without nitrogen fertilizer treatment; 681 and H represented Quanliangyou 681 and Huanghuazhan cultivars, respectively. Proteins with threshold change fold > 2 and *p* value < 0.05 were regarded as up-regulated, while quantitative ratio < 0.5 and *p* value < 0.05 were considered as down-regulated.

**Figure 3 ijms-20-03674-f003:**
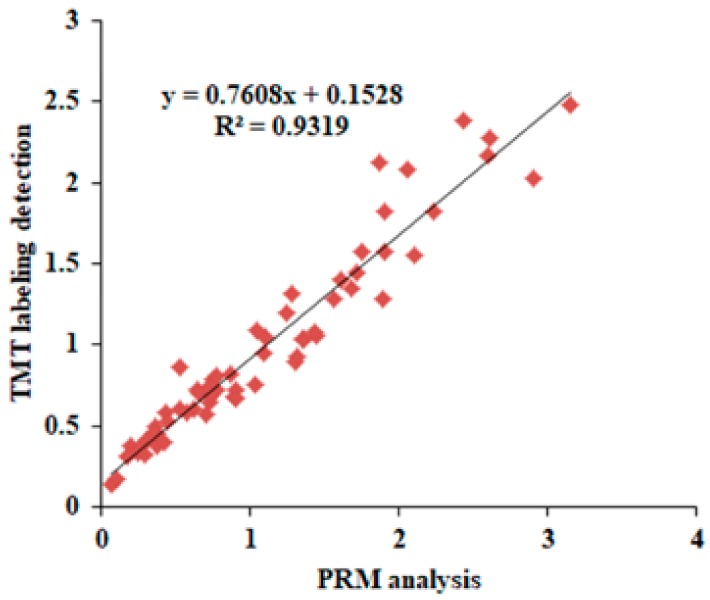
Validation of differentially expressed proteins by parallel reaction monitoring (PRM). Correlation of fold change analyzed by tandem mass tags (TMT) labelling detection platform (*y* axis) with the data obtained by PRM reaction (*x* axis).

**Figure 4 ijms-20-03674-f004:**
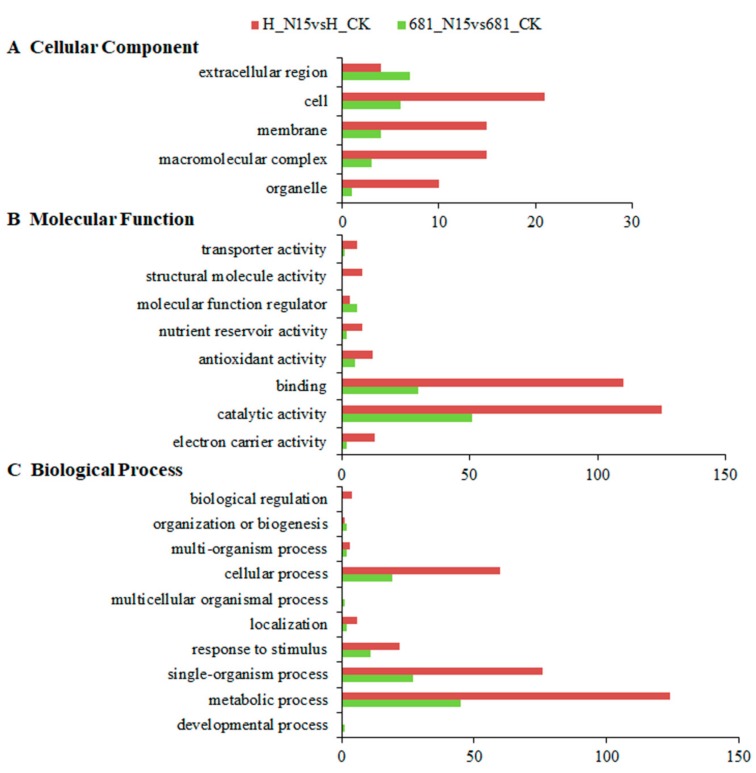
Statistical distribution chart of differentially expressed proteins in the two rice cultivars roots under each Gene Ontology (GO) category (^2nd^ Level). **A.** Cellular Component; **B.** Biological Process; **C.** Molecular Function. N15 represented rice under nitrogen fertilizer treatment with the concentration of while 225 kg/hm^2^, while control check (CK) represented the control without nitrogen fertilizer treatment; 681 and H represented Quanliangyou 681 and Huanghuazhan cultivars, respectively.

**Figure 5 ijms-20-03674-f005:**
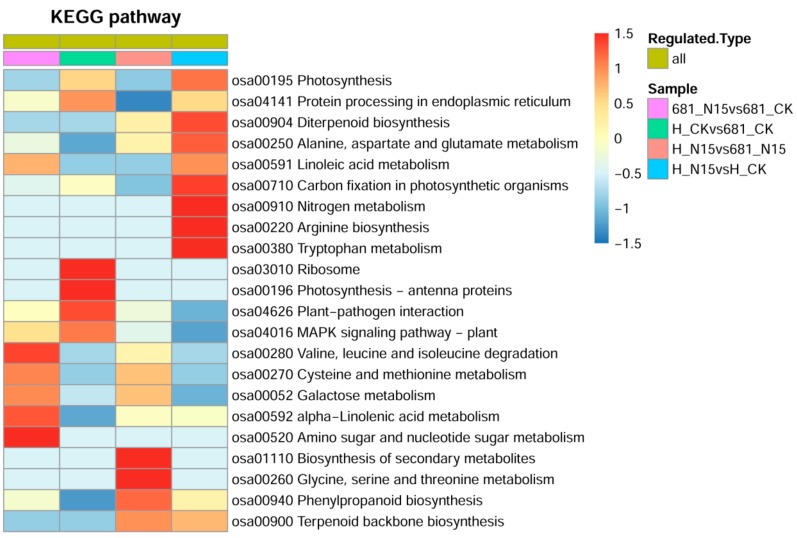
Heatmap for cluster analysis of the enrichment patterns of Encyclopedia of Genes and Genomes (KEGG) pathways. N15 represented rice under nitrogen fertilizer treatment with the concentration of while 225 kg/hm^2^, while control check (CK) represented the control without nitrogen fertilizer treatment; 681 and H represented Quanliangyou 681 and Huanghuazhan cultivars, respectively.

**Table 1 ijms-20-03674-t001:** Dry weight (t/hm²) of rice cultivars under Nitrogen (N) treatment at different levels in different stages.

Cultivar	Level	Tillering Stage	Jointing Stage	Full Heading Stage	Mature Stage
Huanghuazhan	N0	0.57 ± 0.06c	2.34 ± 0.17c	8.70 ± 0.37c	11.86 ± 0.19b
	N9	1.21 ± 0.12b	3.63 ± 0.04b	10.29 ± 0.29c	16.68 ± 0.26a
	N12	1.57 ± 0.15a	4.38 ± 0.17a	12.58 ± 0.45b	17.06 ± 0.35a
	N15	1.78 ± 0.15a	4.61 ± 0.20a	14.94 ± 0.54a	16.86 ± 0.27a
Quanliangyou 681	N0	0.72 ± 0.07b	3.07 ± 0.14d *	9.94 ± 0.46c *	14.77 ± 0.16b *
	N9	2.15 ± 0.02a *	4.25 ± 0.26c *	14.76 ± 0.16b *	18.21 ± 0.29a *
	N12	2.39 ± 0.19a *	4.91 ± 0.07b *	14.59 ± 0.062b *	18.10 ± 0.23a *
	N15	2.48 ± 0.25a *	5.76 ± 0.17a *	16.26 ± 0.13a *	19.23 ± 1.15a *

Note: Huanghuazhan and Quanliangyou 681 rice seedlings were treated with three levels of nitrogen fertilizer (including 135, 180 and 225 kg/hm^2^, labeled as N9, N12, N15, respectively), and the untreated group was considered as the control (labeled as N0). Different letters (a, b, c) indicate significant differences among different nitrogen fertilizer levels for a given cultivar in a given stage based on Duncan’s multiple range tests (*p* < 0.05). Star (*) represents statistical significance between the two cultivars at a given nitrogen fertilizer level in a given stage based on Independent-samples *t* test (*p* < 0.05).

**Table 2 ijms-20-03674-t002:** The differentially expressed proteins involved in ammonium assimilation in the comparisons of H_N15 vs. H_CK, and 681_N15 vs. 681_CK.

Protein Accession	Protein Description	Gene Name	H_N15/H_CK	681_N15/681_CK
B8AQE3	Glutamine synthetase	*OsI_13264*	1.359	3.216
B8AVN6	Glutamine synthetase	*OsI_17756*	0.486	0.782
B8AJN3	Glutamine synthetase	*OsI_10575*	0.259	0.605
B8AZE9	Glutathione synthetase	*OsI_18978*	2.226	1.734
B8AWG6	glutamate synthase 2 [NADH], chloroplastic	*OsI_20922*	0.308	0.605
B8A9D2	probable glutamate carboxypeptidase 2 isoform X1	*OsI_03680*	2.028	1.108
B8AGA6	glutathionyl-hydroquinone reductase YqjG isoform X2	*OsI_06963*	2.115	1.618

Note: In this present study, proteins with the threshold change fold >2 or <0.5, and *p* value < 0.05 were considered as up-regulated and down-regulated proteins, respectively. Black color represented up-regulated protein, and gray color indicated down-regulated protein. N15 represented rice under nitrogen fertilizer treatment with the concentration of 225 kg/hm^2^, while control check (CK) represented the control without N fertilizer treatment; 681 and H represented Quanliangyou 681 and Huanghuazhan cultivar, respectively.

**Table 3 ijms-20-03674-t003:** The differentially expressed proteins involved in signal transduction in the comparisons of H_N15 vs. H_CK, and 681_N15 vs. 681_CK.

Protein Accession	Protein Description	Gene Name	H_N15/H_CK	681_N15/681_CK
A2Z8U1	putative receptor-like protein kinase	*OsI_34134*	3.803	1.294
A2Y361	receptor-like protein kinase FERONIA	*OsI_19449*	2.090	1.301
B8B6Z1	cysteine-rich receptor-like protein kinase 25	*OsI_26338*	1.814	2.017
A2ZC94	1-aminocyclopropane-1-carboxylate oxidase	*OsI_35404*	2.959	3.840
B8AY41	1-aminocyclopropane-1-carboxylate oxidase	*OsI_18467*	2.102	2.676
B8AGJ3	serine/threonine-protein kinase STY46	*OsI_05618*	1.331	2.621
A2XEP1	phosphoenolpyruvate carboxykinase [ATP]	*OsI_10803*	1.220	2.310
B8B2K2	serine/threonine-protein kinase STY46	*OsI_24506*	0.447	0.745
B8ANP3	aspartokinase 1, chloroplastic isoform X1	*OsI_14341*	0.402	0.359

Note: In this present study, proteins with the threshold change fold >2 or <0.5, and *p* value < 0.05 were considered as up-regulated and down-regulated proteins, respectively. Black color represented up-regulated protein, and gray color indicated down-regulated protein. N15 represented rice under nitrogen fertilizer treatment with the concentration of 225 kg/hm^2^, while control check (CK) represented the control without N fertilizer treatment; 681 and H represented Quanliangyou 681 and Huanghuazhan cultivar, respectively.

**Table 4 ijms-20-03674-t004:** The differentially expressed proteins involved in energy metabolism and material transport in the comparisons of H_N15 vs. H_CK, and 681_N15 vs. 681_CK.

Protein Accession	Protein Description	Gene Name	H_N15/H_CK	681_N15/681_CK
Energy Metabolism				
A2Z4G9	probable NAD(P)H-dependent oxidoreductase 1	*OsI_32550*	2.723	1.387
B8BFL3	probable NAD(P)H-dependent oxidoreductase 2	*OsI_32552*	2.242	2.204
A2XNS0	Ferredoxin	*OsI_15810*	2.574	0.622
A2YI62	Ferredoxin--NADP reductase	*OsI_24896*	2.283	1.372
A2ZCK1	Alcohol dehydrogenase family-3	*OSI9Ba083O10*	4.092	1.227
B8ADR5	cytokinin dehydrogenase 2	*OsI_00771*	3.210	1.205
A2YBK1	aldehyde dehydrogenase family 2 member B7, mitochondrial	*OsI_22484*	2.081	1.288
A2XW22	Glutamate dehydrogenase 2, mitochondrial	*GDH2*	1.416	2.212
B8AF09	Glyceraldehyde-3-phosphate dehydrogenase	*OsI_07948*	0.402	0.643
A2Y8B2	L-lactate dehydrogenase	*OsI_21291*	0.371	0.569
A2XU83	Glyceraldehyde-3-phosphate dehydrogenase	*OsI_16160*	0.361	1.098
Material Transport				
B8BLE1	ABC transporter G family member 48	OsI_36727	2.274	1.518
B8B6Q3	ABC transporter G family member 43	OsI_26239	2.152	1.900
A2WPG7	pyrophosphate-energized vacuolar membrane proton pump	OsI_01741	2.025	1.262
A2XAP0	pyrophosphate-energized vacuolar membrane proton pump	OsI_09320	0.390	0.685
B8ACQ3	potassium channel KAT2 isoform X1	OsI_00861	0.486	0.783
B8BBQ9	dipeptide transport ATP-binding protein	OsI_28206	0.445	0.747
A2YP92	"transmembrane protein, putative	OsI_27085	0.432	1.146
A2YVV8	putative copper transporter 5.2	OsI_29464	0.474	0.892
A2YML7	sugar carrier protein C	OsI_26468	1.572	2.925
B8ACJ8	calmodulin-like protein	OsI_01318	1.446	5.881

Note: In this present study, proteins with the threshold change fold >2 or <0.5, and *p* value < 0.05 were considered as up-regulated and down-regulated proteins, respectively. Black color represented up-regulated protein, and gray color indicated down-regulated protein. N15 represented rice under nitrogen fertilizer treatment with the concentration of 225 kg/hm^2^, while control check (CK) represented the control without N fertilizer treatment; 681 and H represented Quanliangyou 681 and Huanghuazhan cultivar, respectively.

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
