# Peer review of "Nitrogen Fertilizer Induced Alterations in The Root Proteome of Two Rice Cultivars"

_ijms, 2019, doi:10.3390/ijms20153674_

Round 1

Reviewer 1 Report

In the manuscript Tang et al describe the effect on the root proteome of three levels of nitrogen fertilizers, Although the results may be of potential interest, my recommendation is to reject in the current form. The English in the manuscript is not of publication quality and require major proof-reading.

Major concerns:

- My major concern is that the study is not hypothesis driven. Introduction presents a concerning scenario about NUE, the overdosage of N fertilisers and its impact on the environment. Then, three levels of fertilizers are compared. However, despite the amount of data collected in the study, there is not a clear conclusion or recommendation arisen from the data. 

- Another major problem, common to most of proteomic based experiments, is to get a clear message out of the amount of data provided by this technique. In my opinion, the number of Figures should be reduced and the discussion should be modified to be clearer. Science is correct, meaningful interpretation of the results is a problem.

 - p value for acceptance of significant changes needs to be corrected to counteract from multitesting (Bonferroni test or similar).

- It is not clear to me how many samples were analysed per group, please, clarify the experimental design including the number of biological replicates analysed.

Minor concerns: 

- Table 2 should be included as supplementary

- Multiple format mistakes as improper use of capitalisation (line 445 Dithiothretiol, Inhibitor).

- Several acronyms are not defined (TEAB line 455, FDR in page 24, PRM in line 499)

- Please change Met by methionine and Cys by cysteine in line 489

- Please, check for grammar mistakes, as:

        line 64 "proteomic approaches has..."

        line 444 "the samples was.."

- 30 m2, 

- kg/hm2

- Figures of poor quality

- Please check for references missing in the list (line 494, Hu et al 2017)

- Capitalization in reference 45 should be omitted

Reviewer 2 Report

Nitrogen use efficiency (NUE) is important topic in reality and scientific interest.  The present study has great results on the new insights into the regulatory mechanisms of nitrogen fertilization in the crop that is very important to identify the potential molecular protein. However, the manuscript should be improved for publication in IJMS.

The authors have to address the following points:

 Abstract should be avoid the suggested results.

In the introduction:

- The Nitrogen fertilizer should be taken into account to Amonia (NH3).

    The author should refer to the paper: Environmental Chemistry 2017, 14 (5), 327-337

- The novelty of this research needs emphasize

3. Results

- Capture of Figure 1 should be in one page.

- The umbers in Fig. 3 A and B are too small to see anything.

- How to evaluate Molecular Function. The comparison with other previously published data is needed.

- So many data in Table 2 that is difficult to follow. What is the differences between the highlights of data.

4. Discussion

- What are enough amounts of GS to incorporate NH4+.

- How to evaluate electron transfer and transmembrane transport.

- Conclusions should separate to other section in which the originality is again emphasized.

5. Materials and Methods.

Protein extraction from rice needs the references. No cited papers are not convincing. 

Protein concentrations determined by BCA must cite at least one reference.

Also, Trypsin digestion should add the cited paper.

Please see the paper related to standard protein: Colloid and Polymer Science 296 (1), 145-155

- Detail of HPLC and LC-MS/MS should be added. The validations of methods are needed.

Round 2

Reviewer 1 Report

The authors have replied to all my questions and the paper has been improved. Thanks for your work.

Reviewer 2 Report

The revised paper was significantly improved by the authors.

Also, the responses are good.

The paper can be accepted in IJMS with current form